



**Opinion: A paradigm shift in investigating the general characteristics of atmospheric new particle formation using field observations**

Markku Kulmala[1,2,3,4], Diego Aliaga[1], Santeri Tuovinen[1], Runlong Cai[1,5], Chao Yan[2,3,4], Federico Bianchi[1], Yafang Cheng[6], Aijun Ding[3,4], Douglas R. Worsnop[1], Tuukka Petäjä[1,3], Katrianne Lehtipalo[1], Pauli Paaasonen[1] and Veli-Matti Kerminen[1,3]

[1] Institute for Atmospheric and Earth System Research/Physics, Faculty of Science, University of Helsinki, 00014 Helsinki, Finland
[2] Aerosol and Haze Laboratory, Beijing Advanced Innovation Center for Soft Matter Science and Engineering, Beijing University of Chemical Technology, 100029 Beijing, China
[3] Joint International research Laboratory of Atmospheric and Earth System Research, School of Atmospheric Sciences, Nanjing University, 210023 Nanjing, China
[4] Nanjing-Helsinki Institute in Atmospheric and Earth System Sciences, Nanjing University, Nanjing, 210023, China.
[5] Shanghai Key Laboratory of Atmospheric Particle Pollution and Prevention (LAP[3]), Department of Environmental Science and Engineering, Jiangwan Campus, Fudan University, 200438 Shanghai, China
[6] Minerva Independent Research Group, Max Planck Institute for Chemistry, 55128 Mainz, Germany

*Correspondence to*: Markku Kulmala (markku.kulmala@helsinki.fi)

**Abstract**

Atmospheric new particle formation (NPF), together with secondary production of particulate matter in the atmosphere, dominate aerosol particle number concentrations and submicron particle mass loads in many environments globally. Our recent investigations show that atmospheric NPF produces a significant amount of particles on days when no clear NPF event has been observed/identified. Furthermore, it has been observed in different environments all around the world that growth rates of nucleation mode particles vary little, usually much less than the measured concentrations of condensable vapors. It has also been observed that the local clustering, which in many cases acts as a starting point of regional new particle formation (NPF), can be described with the formation of intermediate ions at the smallest sizes. These observations, together with a recently developed ranking method, leads us to propose a paradigm shift in atmospheric NPF investigations. In this opinion paper, we will summarize the traditional approach to describe atmospheric NPF, and describe an alternative method, covering both particle formation and initial growth. The opportunities and remaining challenges offered by the new approach are discussed.

**1. Background**

Atmospheric new particle formation (NPF) includes the formation of molecular clusters via different chemical pathways, and the activation of some of these clusters for growth to larger sizes (Zhang et al., 2012; Kulmala et al., 2014; Lee et al., 2019; Kirkby et al., 2023). Depending on their subsequent fate in the atmosphere, essentially whether and how long they will survive from various sink processes (e.g. Kerminen et al., 2004; Pierce and Adams, 2007; Kulmala et al., 2017; Cai et al., 2022), these newly formed particles will contribute to the cloud condensation nuclei in a regional and global atmosphere (Spracklen et al., 2008; Wiedensohler et al., 2009; Kerminen et al., 2012; Gordon et al., 2017; Ren et al., 2021), and



will act as seeds for haze particles during air pollution episodes in urban environments (e.g.
Guo et al., 2014; Kulmala et al., 2022a).
In order to understand how atmospheric NPF influences climate and air quality, and how
these influences have changed over time or will change in the future as a result of
anthropogenic and natural emission changes, we need to have detailed knowledge about
the following issues in different atmospheric environments: 1) what is the general
characteristics of atmospheric NPF, including its frequency and intensity, 2) by which
chemical mechanisms and constituents molecular clusters form and grow to larger sizes, and
3) how effectively newly formed particles reach sizes relevant to climate or air quality. In this
opinion paper, we will focus on the first issue, acknowledging that the synergic effects of all
of them need to be considered in order to get a full understanding of atmospheric NPF. We
concentrate solely on field observations, as the power of laboratory experiments and model
simulations is the strongest when investigating the issues 2 and 3 mentioned above.
Traditionally, the general characteristics of atmospheric NPF have been investigated in terms
of discrete NPF events, during which the formation of new particles has been so intense that
a new mode of particles has been clearly observed. These events take place either locally
close to emission sources of precursor compounds for this phenomenon, or regionally over
distances up to 1000 km or more (Kerminen et al., 2018, and references therein; Chu et al.
2019). With this approach, the particle concentrations resulting from atmospheric NPF can
be quantified only for these clear NPF events, leaving little room for potentially low-intensity
NPF in a regional atmosphere, and providing practically no tools to handle local NPF.
Another problem with the traditional approach is that the subsequent growth of newly
formed particle to sizes relevant to climate or air quality can only be estimated for a small
subset of cases, essentially those when both particle formation and growth take place
relatively homogeneously in the regional atmosphere. In this opinion paper, we will propose
an alternative approach to investigate atmospheric NPF, covering both particle formation
and initial growth on all days with suitable aerosol data. We will discuss the opportunities
that the new approach will offer for future investigations, as well as the remaining
challenges, noting its complementary role when compared with traditional NPF event
analysis and large-scale atmospheric model simulations.

## 2. Approaches to investigate atmospheric NPF using field observations

In this section, we will shortly discuss the approach traditionally used to investigate
atmospheric NPF, including the history leading to this approach and its weaknesses. Based
on our very recent work and findings, we then propose an alternative approach to tackle the
problem, which may lead to a paradigm shift in investigating the general characteristics of
atmospheric new particle formation using field observations. The main features of both
approaches are summarized in Figures 1 and 2, and are discussed in more detail in the
following sections.

### 2.1 Traditional approach and associated shortcomings

Before continuous field observations, our understanding of atmospheric NPF relied entirely
on theories and laboratory experiments. The first steps of NPF were described using classical



nucleation theories which predict a very high dependence of the particle formation rate, or
the nucleation rate, on the concentrations of vapors participating in NPF (e.g. Doyle et al.,
1961; Jaecker-Voirol and Mirabel, 1989; Kulmala et al., 1991; Vehkamäki et al., 2002; Gaman
et al., 2005). For many years, in conducted laboratory experiments, it has been assumed the
binary water-sulfuric acid nucleation to be the only atmospherically relevant NPF pathway,
and early experiments on this system supported the high sensitivity of the nucleation rate to
the gas-phase sulfuric acid concentration (e.g. Wyslouzil et al., 1991; Viisanen et al., 1997).
As a consequence, atmospheric NPF was essentially thought to be an on/off phenomenon
that occurred sporadically under specific atmospheric conditions, essentially at high sulfuric
acid concentrations.
The first field measurements of atmospheric NPF were made for specific types of plumes,
including power plant plumes, in which NPF did not reach a regional extent (see Kerminen et
al., 2018, and references therein). Such measurements were campaign-based, and thus lack
of a statistical view on how frequent and intense NPF was or whether the newly formed
particles were able to grow into sizes relevant to climate or air quality. Later field
observations, based either on campaign-wise or more continuous measurements at fixed
locations, made it possible to identify and characterize regional NPF (Mäkelä et al., 1997;
Kulmala et al., 2004; Kerminen et al., 2018). While such observations have dramatically
enhanced our understanding of atmospheric NPF, they suffered from instrumental
limitations and the non-homogenous nature of air masses that reach the measurement
sites. As a result, it became a common practice to characterize regional NPF by first
estimating the NPF event frequency at the measurement site (i.e. fraction of days showing
clear NPF) using some NPF event classification criteria (e.g. Dal Maso et al., 2005; Kulmala et
al., 2012; Dada et al., 2018), and then determining particle formation and growth rates for a
relatively small sub-set of days (Fig. 1), essentially those being strong and homogenous
enough to permit determination of these quantities (e.g. Nieminen et al., 2018; Chu et al.,
2019; Kanawade et al., 2022).
The traditional approach to investigate and characterize atmospheric NPF based on event
classification has obvious drawbacks. First, there are days when NPF is either relatively weak
or affected by air mass non-homogeneities or changing weather conditions. Such days are
often classified as undefined days (e.g. Dal Maso et al., 2005; Kulmala et al., 2012),
sometimes further divided into a small number of sub-categories (e.g. Buenrostro Mazon et
al., 2009, Dada et al., 2018), or incorrectly classified as non-event days (Kulmala et al.,
2022b). Long-term measurements indicate that these specious days tend to constitute a
large or even a dominant fraction of all the days (Asmi et al., 2011; Kyrö et al., 2014; Dada et
al., 2017; Wang et al., 2017; Kalivitis et al., 2019; Salma and Nemeth, 2019). Further analyses
are thus difficult for such days. Second, the traditional approach is limited in providing
information about the spatial and temporal variability of regional NPF, especially what it
comes to regional intensity. Third, this approach is practically unable to capture sub-regional
scale NPF.
Over the years, the expected on/off behavior of atmospheric NPF has not been borne out by
observations. Since the first simultaneous measurements of NPF and gas-phase sulfuric acid
concentration (Weber et al., 1995, 1996), it has become clear that the observed formation
rate of new particles in the atmosphere often scales between the first and second power of

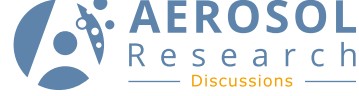

the sulfuric acid concentration (Sihto et al., 2006; Kuang et al., 2008), a much weaker
dependency than predicted by classical nucleation theories discussed above. Such relatively
weak dependence of the particle formation rate on gas-phase concentrations of compounds
participating in NPF have been observed for practically all the NPF pathways identified as
relevant for the atmosphere (Paasonen et al., 2010; Lehtipalo et al., 2018; Yao, et al., 2018;
Brean et al., 2020; He et al., 2021; Yan et al., 2021; Kirkby et al., 2023). Combining these field
and laboratory observations strongly indicates that atmospheric NPF requires additional
vapors beyond sulfuric acid and occurs over much larger concentration ranges of its
precursor compounds than that has been thought before. Kulmala et al. (2022b) developed
a method by which one could detect and even quantify NPF intensity on days traditionally
classified as non-event days. They showed that NPF indeed occurs on such days, and that the
overall contribution of this "quiet NPF" appears to be non-negligible.
Investigations on the growth of newly formed particles to larger sizes are usually based on a
relatively small subset of days on which NPF clearly occurs at a measurement site (Fig. 1).
The main reason for this is that the most commonly used methods for determining particle
growth rates (GR) are applicable only for days during which there is a clear new mode of
particles present in the particle size distribution that can be followed for several hours, and
that the growth is minimally affected by changes in measured air masses (e.g. Dal Maso et
al., 2005; Kulmala et al., 2012). There is a danger that this approach gives a biased view on
GR associated with atmospheric NPF, including the average level, variability and particle size
dependency of GR. Another, very common assumption is that the GR is determined almost
solely by condensation of low-volatile vapors onto newly formed particles. This view is
challenged by the relatively low observed variability of GR in general (e.g. Kerminen et al.,
2018; Nieminen et al., 2018) and, even more importantly, by the weak dependence of the
observed GR on concentrations of "known" low-volatility vapors (Kulmala et al., 2022b;
2023a). Acknowledging that aerosol physical processes (coagulation, cluster collisions with
growing particles) have usually minor influences on GR (Stolzenburg et al., 2023), this
"growth mystery" has two apparent explanations: either 1) our understanding of the mixture
of vapors effectively condensing onto small particles, or the associated thermodynamics, is
fundamentally incomplete, or 2) more volatile atmospheric vapors contribute to GR, e.g. via
chemical reactions on particle surfaces.

**2.2 New, alternative approach and new opportunities**

Instead of separately estimating the frequency of NPF and its intensity for a sub-set of days,
often comprising only a small fraction of all days at any individual site, we propose that
these two quantities will be combined into a probability distribution of the intensity of NPF
that in practice covers all the days (Fig. 2). The intensity of NPF would in this case mean the
formation rates of new particles, $J_C$, at some fixed particle diameter, $d_C$. Provided that the
particle growth rate at sizes close to or slightly above $d_C$ are known, or can be estimated, the
values of $J_C$ can be determined in the same way as in the traditional NPF event analysis (e.g.
Kulmala et al., 2012), and then integrated over a desired period of time (daily, sub-daily or
instantaneous values). To get the best benefit of the derived distribution of $J_C$ for subsequent
application purposes, the chosen value of $d_C$ should be large enough so that the complicated
and poorly understood processes that determine survival probabilities of growing clusters
and particles would mainly be restricted to the sizes below $d_C$ (see, e.g. Kulmala et al., 2017).



According to our current understanding in this respect (Cai et al., 2022; Tuovinen et al.,
2022), $d_C$ should be 3 nm in minimum, and preferably somewhat larger under heavily
polluted conditions. The upper limit of $d_C$ should be selected so that the calculated values of
$J_C$ would be minimally affected by primary particle sources. While this is of little concern in
remote or most rural areas, fresh primary particles, e.g., from traffic emission, are known to
extend size well below 10 nm (e.g. Rönkkö et al., 2017).
An immediate question is how to determine the probability distribution of the particle
formation rate, especially at the lower end of this distribution that represents weak to
moderate NPF. Kulmala et al. (2022b) demonstrated that by averaging and suitably scaling
over a large number of measurement days, it is possible to estimate particle formation rates
on days previously classified as non-event days using the traditional NPF event classification
methods. However, this is not the only available option. Previous analyses have shown that
atmospheric NPF is strongly associated with concentrations of intermediate ions, i.e., ions in
the size range from 2 to a few nm (e.g. Tammet et al., 2014; Leino et al., 2016). Motivated by
this finding, we recently investigated the sensitivity of NPF to the total particle number
concentration in a similar size range using long-term measurement data from the SMEAR II
station in Finland (Aliaga et al., 2023). We found that the days with higher (suitably scaled)
2.5-5 nm particle concentrations (high ranking) showed, on average, both higher particle
formation rates and, in terms of traditional NPF event classification, higher probability of a
NPF event to occur (Aliaga et al., 2023). Such a ranking method appears a promising
candidate for creating a probability distribution of particle formation rates; however, its
performance in different environments needs to be carefully tested. So far, besides the
SMEAR II station, the preliminary results from a mountain site (Chacaltaya in Bolivia) and
polluted sites (Beijing in China, Po Valley in Italy) are promising.
Concerning particle growth, we propose that rather than determining GR for only a small
subset of days, as usually done when analyzing field measurements, one should aim to find a
relation between GR and the prevailing chemical environment. By a chemical environment
we mean the presence (concentrations) of vapors that potentially contribute to GR, and the
activity of processes (condensation, heterogeneous reactions) that link these vapors to GR.
Important to keep in mind when doing all this that not only the least volatile vapors, but also
more volatile vapors capable of producing non-volatile vapors via heterogeneous reactions
in and/or on particles, may have a significant contribution to GR. To a first approximation,
regionally representative values of GR and its variability could be derived using the largest
sub-set of high rankings that display particle growth; even without detailed information
needed to tie GR with the chemical environment. This approach can be justified by the fact
that GR is determined by the prevailing chemical environment rather than the intensity of
NPF, so that losing information from days with low-intensity NPF does not cause a serious
bias in GR estimates (Kulmala et al., 2022b). It is also possible to improve the
representativity of the GR for individual days, depending on the length of the observation
data set, by investigating the typical GR in different seasons, under different meteorological
conditions or under otherwise varying situations in the chemical environment.
Analysis of both ions and particles in the 2-5 nm size range might provide a tool to combine
regional and sub-regional NPF into the same framework. By using measurement data from
the SMEAR II station and performing a theoretical analysis, it was demonstrated that the



concentration of negative ions in a narrow size range of 2.0-2.3 nm could be related to the
intensity of NPF averaged over a spatial scale of the order 1 km surrounding the
measurement site (Kulmala et al., 2023b; Tuovinen et al., 2023). If this is more generally
applicable, including other sites with differing molecular particle forming mechanisms,
targeted measurements of 2.0-2.3 nm ions could thereby be applied for identifying, and
possibly quantifying, how effectively a specific (local) environment will produce new
particles into the atmosphere (Fig. 2). Although such measurements say nothing about the
subsequent fate of these particles, to a first approximation we may assume that they will
grow essentially in the same manner as any newly formed in the same regional atmosphere
or, more specifically, in the same prevailing chemical environment mentioned above. We
cannot go to sizes smaller than 2 nm, since concentrations and dynamics of smaller ions are
determined by processes which have very little to do with NPF (e.g. Tammet et al., 2014).
**3. Paradigm shift and remaining challenges**
Based on these recent results and the new reasoning presented above, we suggest the
following paradigm shift when investigating the general characteristics of atmospheric NPF
using field measurements (see also Fig. 2):
1) Instead of making binary (event, non-event) classification of NPF, we will utilize all
days in the analysis and use a more continuous approach, such as the ranking
method, for statistical information on the intensity of NPF.
2) We use particle and ion number concentrations in the smallest possible size regimes:
a. total particles (2.5-5 nm) or intermediate ions (2-7 nm) to study regional NPF;
b. ions at diameters as close to 2 nm as possible to study local NPF.
3) We use the regionally representative particle growth rates, derived from the largest
possibly subset of data, to calculate:
a. regional values of particle formation rates at selected sizes (3-5 nm) and
integrated over desired time periods (instantaneous to daily);
b. local particle formation rates over selected areas, and their relative
contributions to regional NPF.
4) The particle formation rates can be determined for all days, and its distribution can
be given as a continuous function of different parameters.
The main advance over the traditional method is that the new paradigm provides estimates
of particle formation rates for all measurement days, and in principle even continuous values
as a function of time. But it remains to be investigated what the best time resolution is for
doing this analysis in practice. This, together with regionally representative particle growth
rates, provides us with a tool to quantify the contributions of both local and regional NPF to
total particle number concentrations in a regional atmosphere.
Despite its highly promising potential to investigate atmospheric NPF, the new paradigm
faces apparent challenges as well. For example, while continuous aerosol size distribution
measurements are being conducted in tens of locations worldwide (e.g. Rose et al., 2020), a
dominant fraction of these sites do not currently have proper instrumentation (e.g. NAIS,
PSM, nano-DMPS) for measuring the sub-5 nm size range needed for applying the



alternative approach introduced here. The lack of available measurement is even more
severe for ion measurements necessary for determining local or sub-regional NPF.
Dealing with primary emissions has been found to be difficult when investigating regional
NPF in polluted environments (e.g. Woo et al., 2001; Ahlm et al., 2012; Nemeth et al., 2018;
Pushpawela et al., 2018; Zhou et al., 2020; Kanawade et al., 2022), and the new, alternative
approach introduced here is not expected to be free from the influences of primary
emissions. Recently, methods have been developed to estimate, and potentially exclude,
primary emissions originating from emissions, e.g., traffic (Okuljar et al., 2021; Chen et al.,
2023), but the suitability of these methods for this proposed new approach remains to be
investigated.
Particle survival probabilities are sensitive to the combined effect of the degree of pollution
and particle growth rate, and there are large uncertainties in predicting this quantity in the
sub-3 to 5 nm size range, especially in polluted environments (e.g. Kulmala et al., 2017; Cai
et al., 2022; Tuovinen et al., 2022). This feature does not cause a major problem for
investigating regional NPF, as long as the probability distribution of the particle formation
rate is derived at large enough sizes, preferably at 5 nm and in minimum at 3 nm.
Concerning sub-regional NPF, we need to investigate whether and how particle survival
probabilities influence the connection between ion concentrations close to 2 nm and
particles formation rates at larger sizes, as this connection is expected to depend on the
environment and prevailing conditions.
It is well known that both the occurrence and intensity of NPF vary seasonally at most of the
sites (Dall'Osto et al., 2018; Kerminen et al., 2018; Nieminen et al., 2018; Chu et al., 2019,
Brean et al., 2023). The potentially large seasonal variability of the mixture and
concentrations of vapors contributing particle formation and growth needs to be kept in
mind when calculating particle formation and growth rates, and when determining
representative distributions for these quantities.
Finally, although the tools introduced here provide a first-order estimate on particle growth
rates in different environments, we are far from a full understanding on which vapors and
processes determine GR in different environmental conditions. As a result, much future work
is needed to define, characterize, and quantify the chemical regimes and processes that
eventually determine GR and its variability, and how this variability feeds back into
estimating particle formation rates during low-intensity NPF.
**4. Conclusions**
In this opinion, we have proposed a new method/approach and elucidated a paradigm shift
in investigating atmospheric NPF using field observations. Contrary to the traditional event-
based classification of individual days, the new approach looks at atmospheric NPF in a more
statistical sense, aiming to create a probability distribution of particle formation and growth
rates for all the days from continuous measurements at individual sites. While generally
applicable to regional NPF, we also present ideas on how this same framework could be
extended to sub-regional, or local, NPF.



The new approach provides a method to quantitatively estimate the contribution of
atmospheric NPF to particle number concentration budgets in a regional atmosphere. If
supported by additional measurements in areas with distinct sources for NPF precursors, the
relative contributions of such source areas to the regional NPF can, in principle, be
estimated. The results from the new approach can be extended to continental scales,
provided that continuous measurement data from different representative regions are
available.
The approach proposed here should be thought as complementary to the traditional NPF
event analysis and large-scale model simulations. The traditional NPF event analysis has
been widely used in the past, so its application to an entirely new data set offers a simple
way to get idea on how important NPF is in that particular environment, and how it
compares to other environments investigated earlier. The traditional NPF event analysis
remains to be a powerful tool to select cases (days) for some special investigation purposes,
such as investigating atmospheric NPF pathways and associated precursor chemistry
associated with atmospheric NPF. The large-scale view on atmospheric NPF, including its
climatic and health effects, as well as the associated feedback processes, has relied almost
entirely on model simulations in the past. The proposed approach brings atmospheric
measurements on NPF closer to results from large-scale model simulations and, at the very
least, the new paradigm offers an improved way to utilize measurement data to constrain
and evaluate models simulating atmospheric NPF.

**Author contribution**
Markku Kulmala and Veli-Matti Kerminen had the original idea for opinion paper. YC, AD
and DRW contributed to developing the idea further. VMK, MK, KL and PP wrote the first
version of the paper.  DA, ST, RC, CY, FB, TP, VMK and MK developed the material and
results behind the opinion paper. All coauthors contributed the final version of the paper.

**Competing interests**
Markku Kulmala is a member of the editorial board of Aerosol Research.

**Acknowledgements**
We acknowledge the following projects: ACCC Flagship funded by the Academy of Finland
grant number 337549, Academy professorship funded by the Academy of Finland  (grant no.
302958), Academy of Finland projects no. 325656, 311932, 334792, 316114, 325647,
325681, 333397, 328616, 357902, 345510, 347782, "Quantifying carbon sink, CarbonSink+
and their interaction with air quality" INAR project funded by Jane and Aatos Erkko
Foundation, "Gigacity" project funded by Wihuri foundation,  European Research Council
(ERC) project ATM-GTP Contract No. 742206, and European Union via Non-CO2 Forcers and
their Climate, Weather, Air Quality and Health Impacts (FOCI), and CRiceS (No 101003826),
RI-URBANS (101036245), EMME-CARE (856612) and FORCeS (821205). University of Helsinki
support via ACTRIS-HY is acknowledged. Support of the technical and scientific staff in
Hyytiälä and BUCT/AHL are acknowledged.



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

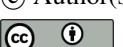



**Figures**

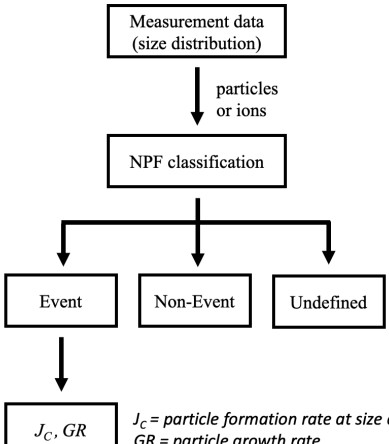

Figure 1.   Schematics of the traditional method used to characterize regional atmospheric
NPF.

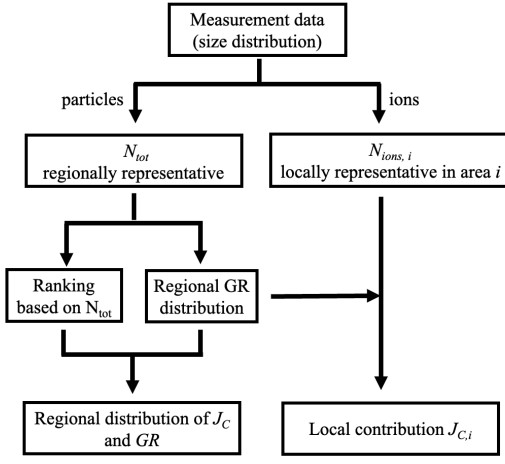

Figure 2.  Schematics of the new method proposed in this paper to characterize both
regional and local NPF.