# Peer review of "Opinion: A paradigm shift in investigating the general characteristics of atmospheric new particle formation using field observations"

_Aerosol Research, 2023_

## Author Comment (AC1)

ANSWERS TO REFEREES' comments

M. Kulmala et al.

REF 1

*In this MS the authors propose a new way to study atmospheric NPF, in a more statistical sense. Starting from data collected in field observations, this method aims to create a probability distribution of particle formation (JC) and growth rates (GR) to obtain statistical information on the intensity of NPF. Instead of making the traditional classification of NPF, between event and non-event, and analysing only a sub-set of data, this method uses a continuous approach by analysing all available days. The analysis is performed on particle and ion number concentrations in the size range of particles (2.5-5 nm) or intermediate ions (2-7 nm). Although this is an innovative method that can quantify both local and regional NPF contributions to the total particle number concentrations in the atmosphere, it is very limited. It can only be used if the measurement site has adequate instrumentation to measure the sub-5 nm size range needed to apply the method.*

We thank the reviewer for the very good overall statement. The reason for limitation is that if we want to investigate NPF, we need instruments that can directly measure the recently formed nanoparticles. If the cut size is 5 nm or bigger, we are able to catch only the growth of particles, not the NPF process itself, which hinders our ability study its mechanism, occurrence etc. However, if the cut size is 5 nm or bigger, instead of NPF mechanism, our approach can then be applied to investigate the growth of particles.

*Therefore, the proposed approach can represent a useful but complementary tool to the traditional NPF event analysis.*

*I think that this is an important work since NPF is a topic of current interest and the authors are well-informed and experienced about it. Furthermore, this work is well written, offering a clear exposition of the topic. Therefore, this manuscript has enough quality to be published in this journal.*

*I just have few suggestions to the authors:*

*Line 25: the first sentence of the abstract "Atmospheric new particle formation (NPF), together with secondary production", seems to mean that the NPF is not from secondary origin.*

That was not our purpose. We modified the sentence into the following form: "Atmospheric new particle formation (NPF) and associated production of secondary particulate matter dominate aerosol particle number concentrations and submicron particle mass loadings in many environments globally."

*Line 66: it is not clear "in terms of discrete".*

We modified the text to clarify this issue: "Traditionally, the general characteristics of atmospheric NPF have been investigated by concentrating on so-called NPF events, during

which the formation of new particles has been so intense that a new mode of particles has been clearly observed."

*Line 112: Please explain "sizes relevant to climate or air quality".*

We added: "…, essentially particles larger than about 50-100 nm in diameter"

**Citation**: https://doi.org/10.5194/ar-2023-19-RC1

REF 2:

*In this manuscript, the authors elaborate on a novel approach to describe new particle formation (NPF) in the Earth's atmosphere. This method complements traditional NPF analysis rather than replacing it. The use of a continuous approach, as opposed to event-based analysis, may serve as a powerful tool to quantify the contribution of NPF phenomena to atmospheric aerosol number concentrations and submicron mass, as well as to evaluate model simulations of NPF.*

We are thankful to the reviewer for the very nice overall statement, and for the comments below.

*Therefore, it is strongly recommended that this article be published in AR. Please find below a few minor comments for your consideration:*

*Line 155: Not all readers may be familiar to the concept of "quiet NPF" and it is recommended to briefly describe the term.*

We modified the text into the following form:

"Kulmala et al. (2022b) developed a method by which one could detect and even quantify the intensity of NPF on days traditionally classified as non-event days. They showed that NPF indeed occurs also on such days, and termed it "quiet NPF" because this phenomenon does not produce a visible NPF event in a picture illustrating the time evolution of a particle number size distribution over a single day. The overall contribution of "quiet NPF" to the total production of new atmospheric aerosol particles appears to be non-negligible (Kulmala et al., 2022)."

*Line 167: Is it meant that low variability is observed worldwide? If so, it is worth providing an indicative average value or clarify it.*

We clarified this issue by modifying the text into the following form:

"This view is challenged by two things. First, the average values of GR have been observed to vary little, often less than a factor of 2-3, between different environments, and even less between different sites within a certain type of an environment (e.g. Kerminen et al., 2018; Nieminen et al., 2018). Second, and perhaps more importantly, GR was found to depend rather weakly on the concentrations of "known" low-volatility vapors at two entirely different sites: urban Beijing, China (Kulmala et al., 2022a) and SMEAR II station in boreal forest, Finland (Kulmala, 2023)."

*Line 175: An appropriate reference is needed here.*

We added: "(see detailed discussion in Kulmala et al., 2022a)".

*Line 196: "to extend in size well below 10 nm".*

Corrected.

*Line 211: A brief description of the ranking method will be useful.*

A very brief description of the ranking method has already been presented in the few previous lines prior to line 211. We feel that a more detailed description in this method in this paper is unnecessary, especially because the paper providing a very detailed description of the method has now been published (see updated reference information of Aliaga et al., 2023). We slightly modified the current description of the ranking method to make the text a bit more easier to digest for readers not willing to look at the paper by Aliaga et al. (2023)

*Line 219: Provide a specific example to illustrate a typical chemical environment, such as the boreal environment. Identify the species that may be found at varying levels of volatility.*

The whole idea of introducing the concept "chemical environment" is that GR is (much) more tightly connected to a chemical environment than to any specific physical environment. Since this appeared to remain unclear based on our original text, we added the following sentence into the revised manuscript:

"We note that in any physical environment, its chemical environment determining GR may vary with time due to the variability in the ambient temperature, solar radiation and relatively humidity, or due to changes in anthropogenic or biogenic emissions affecting this environment."

*Line 224: An appropriate reference is needed here.*

Reference to Stolzenburg et al. (2023) was added.

Stolzenburg, D., Cai, R., Blichner, S., Kontkanen, J., Zhou, P., Makkonen, R., Kerminen, V.-M., Kulmala, M., Riipinen, I., and Kangasluoma, J.: Atmospheric nanoparticle growth, Rev. Mod. Phys., 95, 045002, 2023.

*Line 305: This is too generic, name typical variables that have an impact on this connection.*

We modified this sentence into the following form:

"Concerning sub-regional NPF, we need to investigate whether and how particle survival probabilities influence the connection between ion concentrations close to 2 nm and particles formation rates at larger sizes, more specifically how the survival of sub-5 nm particles depends on their growth rate and background particle loading (condensation sink) in the considered environment (e.g. Tuovinen et al., 2022)."

**Citation**: https://doi.org/10.5194/ar-2023-19-RC2